# Smart Vision-Language Reasoners

Denisa Roberts [1]   Lucas Roberts [2]

## Abstract

In this article, we investigate vision-language models (VLM) as reasoners. The ability to form abstractions underlies mathematical reasoning, problem-solving, and other Math AI tasks. Several formalisms have been given to these underlying abstractions and skills utilized by humans and intelligent systems for reasoning. Furthermore, human reasoning is inherently multimodal, and as such, we focus our investigations on multimodal AI. In this article, we employ the abstractions given in the SMART task (Simple Multimodal Algorithmic Reasoning Task) introduced in (Cherian et al., 2022) as meta-reasoning and problem-solving skills along eight axes: math, counting, path, measure, logic, spatial, and pattern. We investigate the ability of vision-language models to reason along these axes and seek avenues of improvement. Including composite representations with vision-language cross-attention enabled learning multimodal representations adaptively from fused frozen pretrained backbones for better visual grounding. Furthermore, proper hyperparameter and other training choices led to strong improvements (up to $48\%$ gain in accuracy) on the SMART task, further underscoring the power of deep multimodal learning. The smartest VLM, which includes a novel QF multimodal layer, improves upon the best previous baselines in every one of the eight fundamental reasoning skills. End-to-end code is available at github.com/smarter-vlm/smarter.

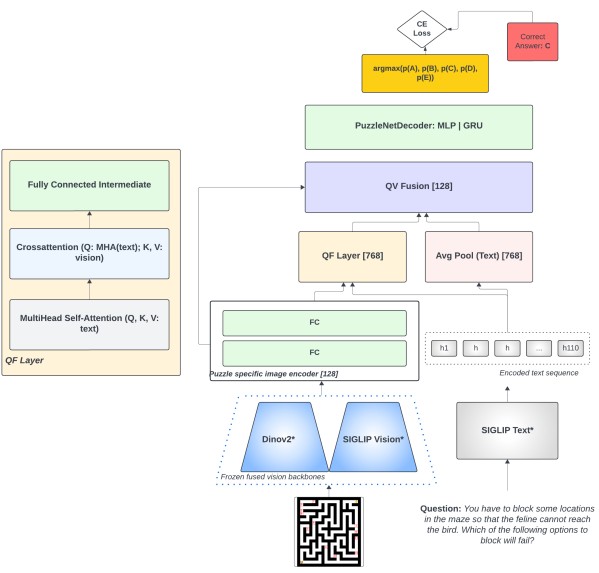

*Figure 1.* The smarterVLM reasoner architecture (right) and the novel QF layer (left). Vision (DinoV2+SigLIP) and language (SigLIP) backbones are frozen. All other layers are trained from scratch.

## 1. Introduction

Human intelligence is oftentimes associated with the ability to operate on mathematical abstractions. In (Chollet, 2019) the author conducts an in depth discussion and formulates

a formal definition of intelligence based on algorithmic information theory. Several meta characteristics of intelligent systems are listed as scope, generalization difficulty, priors and experience. On a different but related axis, (Didolkar et al., 2024) speaks of metacognitive capabilities of large language models, abilities that underlie all problem solving, including math problems. In a related work in the multimodal domain (Cherian et al., 2022), a Simple Multimodal Algorithmic Reasoning Task (SMART) is introduced with visual-linguistic puzzles designed for children in the 6-8 age group (the US Kangaroo Olympiad style). In this work, an explicit categorization of underlying skills utilized by humans in problem solving are labeled and tallied as they get employed in solving puzzles as measure, path, pattern, logic, math, algebra, and spatial skills. Furthermore, reasoning must be multimodal because humans have multiple senses whose inputs are amalgamated to reason at higher abstractions. Better abstractions are akin to better mental representations. Deep neural networks excel at learning

[1]Department of Computer Science, New York University, New York, USA. [2]Yext Inc., New York, USA. Correspondence to: Denisa Roberts <dao9853@nyu.edu>.

*The first AI for MATH Workshop at the 41st International Conference on Machine Learning*, Vienna, Austria. Copyright 2024 by the author(s).

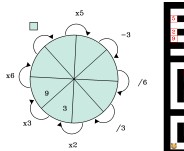 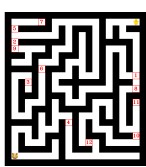 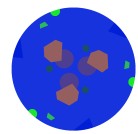 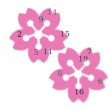 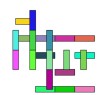 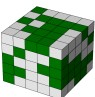 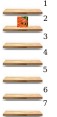 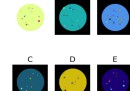

*Figure 2.* **Math Question**: *What do we need to put in the square to get a correct diagram?* **Answer Options:** A: -3; **B**: /9; C: x6; D: x2; E: 2; **Path Question with Sequence Answer**: *You have to block some locations in the maze so that the feline cannot reach the bird. Which of the following options to block will fail?* **Answer Options:** A: 1, 2, and 3; **B**: 4; **C**: 5, 6, and 7; D: 8 and 9 ; E: 10, 11, and 12. **Counting Question**: *The entire pie is divided among several children. Each child receives a piece of pie, and each piece of pie looks identical. The maximum possible number of children there is:* **Answer Options:** A: 7; B: 2; C: 1; D: 4; **E**: 3. **Algebra Question**: *The entire pie is divided among several children. Each child receives a piece of pie, and each piece of pie looks identical. The maximum possible number of children there is:* **Answer Options:** A: 5; B: 4; C: 2; **D**: 0; E: 6. **Measure Question**:*A student had a few canes with a height of 1 cm and a length of 5 cm. Using the canes, she built the arrangement illustrated. What is the width of the arrangement?* **Answer Options: A**: 20; B: 30; C: 15; D: 5; E: 35. **Spatial Question**: *Cristina made a setup using some green blocks and 94 white blocks. How many of these white blocks are not visible in the figure?* **Answer Options:** A: 28; B: 61; **C**: 64; D: 90; E: 79. **Logic Question**: *Emily has 7 toy items: a remote, a hair brush, a truck, an eraser, a rubber duck, carrots, and a toe ring. She keeps each toy at a different row of the shelf. The carrots lower to toe ring. Remote lower to truck and toe ring higher to truck. Toe ring higher to rubber duck. She keeps carrots as shown. On which row can the rubber duck not be placed?* **Answer Options:** A: 4; B: 3; **C**: 7; D: 5; E: 6. **Pattern Question**: *Which picture on the right matches with the left, if we invert the colors?* **Answer Options:** A; B; C; D; **E**.

(artificial) representations (Bengio et al., 2013).

We base our investigations in this article on a few **conjectures** with respect to **mathematical reasoning and general problem-solving**:

- Intelligence is related to **multimodal reasoning**. If a person is deaf and cannot hear more than 50% of what is being said, the speech modality input is supplemented with reading faces and other visual aids (vision modalities), captions (text), as well as other modalities (all the other senses, as well as enhanced reasoning and computation abilities). Therefore, it is worth striving to improve multimodal deep learning architectures and their reasoning abilities.

- **Training** leads to learning and improved reasoning. Math and physics Olympiad expert competitors practice more than non-experts in environments that value the pursuit. Taking a classic example, the Polgar sisters-three chess grandmasters-were trained to excel at chess. Similarly in the musical domain, Mozart and Beethoven were trained from a young age to excel in music. In fact, one interpretation of evolution is the act of training and learning. We learned to grow better brains out of necessity. Therefore, training neural networks specifically to enhance reasoning makes evolutionary sense.

- Intelligence is related to better **abstractions** and those are related to better **representations**. Expert chess players, thespians, martial artists, mathematicians, and coders all develop fine-grained relevant mental representations so they can better reason and imagine new ideas rapidly. Therefore, improving the representations

derived with deep learning architectures is worth pursuing (better image, text etc. representations as well as their cross-play).

- If we make neural networks better at **reasoning** (we include here a few types of reasoning such as creative problem-solving in math, physics, logic and coding algorithms, puzzles and IQ tests, learning, planning and decision making), these skills may be **transferable** to science, strategy, medicine, law, and commonsense, with far reaching real world impact.

## 2. Related Work

**Reasoning** Surveys of deep learning for mathematical reasoning such as (Lu et al., 2022; Sun et al., 2023) mentioned the relatively smaller subset of works on multimodal / vision-language models in this space, with datasets and models which are smaller, niche, and mostly using visual question answering frameworks. These approaches are lacking since they are trained on natural images and not on models trained on vision *and language datasets*. Subsequent works such as (Zhang et al., 2024b; Wu et al., 2024; Lu et al., 2023) and this article, aim to enrich the multimodal mathematical reasoning domain.

**Vision-Language Models** Opportunities for improvement of vision language models still exist along problem-solving and algorithmic reasoning ability, visual grounding, as well as architectures for encoding, decoding and aligning (Karamcheti et al., 2024; Tong et al., 2024; Liu et al., 2023; Wu & Xie, 2023). In this article we focus on the reasoning ability along eight dimensions of reasoning. In the reasoning realm, much recent work focuses on evaluating vision-language models on general multimodal tasks (Yue

et al., 2023; Lu et al., 2023) or applying a chain of thought approach for in context learning (Zhang et al., 2023; 2024a). In (Azerbayev et al., 2023) large language models are pretrained to solve text only questions bringing on the full power of heavy-weight LLM, but without taking the visual signal into account. Then a separate line of work builds very large general Vision-Language Models (VLM) akin to an LLM. In (Li et al., 2023b) a vision-language architecture, the Query Transformer, adds transformer layers to frozen image and text encoders and learns in a contrastive pretraining paradigm on massive datasets. Llava (Liu et al., 2024a; 2023) versions (Liu et al., 2023) emerge as a multimodal instruction tuned large model finetuned on a science dataset. In (Gao et al., 2023) authors use an LLM and parameter-efficient visual instruction tuning, focusing on learning efficiently only the adapters, with early fusion of visual tokens in LLM layers. In (Tong et al., 2024) the deficiencies in visual grounding of large multimodal models is investigated and mixtures of visual features are proposed to improve vision modality. In (Li et al., 2023a), another pretrained and visual instruction tuned framework is proposed, employing CLIP (Radford et al., 2021), Llama (Touvron et al., 2023) and Perceiver adapters (from Flamingo) (Alayrac et al., 2022) as well as a dataset, MIMIC-IT. The pretrained vision encoder in DinoV2 (Oquab et al., 2023) aims to leverage different techniques and diversity of images to pretrain backbones for the purpose of using them as general-purpose features. However, it is not necessarily true that even this general purpose encoder will encode all the visual signals, so fusing with representations from a backbone pretrained in a different fashion, for instance the SigLIP (Zhai et al., 2023) representation, may provide additional visual signal boost. Specifically, SigLIP improves on CLIP (Radford et al., 2021) for language-image pretraining by employing a sigmoid loss instead of the constrastive learning with softmax normalization and performs better across tasks.

**Multimodality and representation learning in Math AI**. MATH-AI research has focused primarily on language models, but problem solving is inherently multimodal (text, image, diagrams, tables, numbers, symbols). In the current article we utilize the vision and text modalities to encode all concepts included in each puzzle, without employing separate encodings and/or representations for symbols or numbers. Multiple works focus on representation learning and reasoning with more than the specialized mathematical visual modality, such as symbols and numbers on symbolic reasoning (Li et al., 2023c), specialized representation learning works for numbers (Golkar et al., 2023), as well as more complex hierarchical math concepts with graphs in (Rute et al., 2024), which would be interesting to include as further modalities in future works. In a related vein, (Wu & Xie, 2023) and (Tong et al., 2024), proffer the reasoning ca-

pabilities of multimodal large language models and explores their visual representation learning abilities.

## 2.1. Benchmark, Dataset, and Challenges

So how can we help (deep) artificial neural networks reason better? In (Cherian et al., 2022) experiments show that the visual signal is very important in solving complex multi-reasoning skill puzzles and, despite being very large, language-only models lag behind visual language models in terms of performance. Conversely, in (Zhang et al., 2024b) the conclusion appears to be that large multimodal models cannot truly understand the visual diagrams for mathematical reasoning, along the line of weak visual grounding and poor attention to visual detail in (Tong et al., 2024) and (Wu & Xie, 2023) for large multimodal models for math, question answering, and other reasoning tasks. The Simple Multimodal Algorithmic Reasoning Task (SMART) introduced in (Cherian et al., 2022) contains puzzles that measure intelligence across eight different reasoning skill classes: counting, math, logic, path, measure, logic, and pattern. Problems include an image and a text question and are formulated as multiple choice. We can see a few examples of problems in Figure 2. Baseline models trained in (Cherian et al., 2022) struggle to solve this task, especially when employing transformers. In the past, specialized neural networks such as (Mikuła et al., 2023) have been developed to solve specific reasoning tasks, specifically premise selection in automated theorem proving. In this article, we investigate how we can craft and train deep neural networks which employ several types of deep learning blocks and multimodal inputs from deep frozen transformers to reason better across the eight meta reasoning axes in the SMART task.

**The SMART reasoning task and baselines**. A set of vision-language models are trained as benchmarks in (Cherian et al., 2022) and SMART-101 with 202K text-image pairs for train, validation and test dataset is released. There are 101 origin puzzles, and additional problems are generated programatically in each puzzle group for a total of 202,000 question-image pairs. Figure 2 clearly describes a training example problem. All trained VLMs struggle on the SMART task, with transformers underperforming ResNet50 (He et al., 2016) based models. The learning tasks depend on the type of puzzle and are in the classification, regression, and sequence generation category. Several image and text encoder backbones are considered. A puzzle specific set of image features are learned via an MLP and the text embeddings are aggregated using an LSTM layer. The decoder for sequence generation is another LSTM layer. All image encoders are finetuned. Based on these characteristics, there are a few research opportunities worth exploring, especially since transformer-based VLM reasoners are doing so poorly on the challenging SMART task.

*Table 1.* Skill class accuracy for original baselines with a 10hr budget training. All backbones are frozen unless noted otherwise.

| SMART BASELINE | COUNTING | MATH | LOGIC | PATH | ALGEBRA | MEASURE | SPATIAL | PATTERN | OVERALL |
|---|---|---|---|---|---|---|---|---|---|
| BERT+RESNET50 | 35.6 | **26.4** | 36.8 | 21.5 | 18.1 | 26.0 | 32.2 | 27.0 | 28.0 |
| BERT+RESNET50(UNFROZEN) | 35.7 | 20.8 | 39.6 | 22.2 | 18.4 | 28.2 | 33.7 | 30.6 | 28.2 |
| BERT+MAE (HE ET AL., 2022) | 29.8 | 19.7 | 29.4 | 20.5 | 16.1 | 18.9 | 26.6 | 27.8 | 23.1 |
| CLIP VLM | 35.5 | 8.6 | 27.1 | 17.9 | 11.8 | 16.0 | 26.8 | 26.3 | 22 |

*Table 2.* Validation Accuracy per Skill Class (counting, math, logic, path) per Architectural, Optimization and Hyperparameter Choices. The fused vision encoder is DinoV2+SigLIP. From CometML multimodalAI.

| CHOICES | COUNTING | MATH | LOGIC | PATH | VISION | LANGUAGE |
|---|---|---|---|---|---|---|
| BASELINE: RESNET50+MBERT | 23.4 | 8.1 | 18.9 | 17.9 | RESNET50 | MBERT |
| BASELINE: RESNET50+BERT | 23.4 | 8.1 | 19.2 | 17.8 | RESNET50 | BERT |
| LSTM DECODER SIGLIP VISION | 24.6 | 7.9 | 17.9 | 17.9 | SIGLIP | SIGLIP |
| LSTM DECODER WITH FUSED VISION | 27.7 | 8.4 | 21.3 | 18.6 | FUSED | SIGLIP |
| NON-ADAPTIVE IMAGE REPRESENTATION | 27.2 | 8.4 | 20.2 | 18.5 | FUSED | SIGLIP |
| EXTRA RESIDUAL CONNECTION IN MLP DECODER | 21.5 | 7.4 | 17.5 | 17.8 | FUSED | SIGLIP |
| WARMUP STEPS 0 | 29.8 | 7.4 | 21.3 | 20.3 | FUSED | SIGLIP |
| WARMUP STEPS 0.06 PERCENT | 30 | 7.8 | 22.3 | 19 | FUSED | SIGLIP |
| WARMUP STEPS 0.01 PERCENT NO EXTRA RESIDUALS | 29.6 | 8.5 | 22.9 | 19.3 | FUSED | SIGLIP |
| 10 WARMUP STEPS | 29.9 | 8.1 | 17.8 | 18.8 | FUSED | SIGLIP |
| BATCH SIZE 64 | 26.1 | 8.2 | 21.4 | 18.8 | FUSED | SIGLIP |
| ADAPTIVE IMAGE REPR SIZE 256 | 25.3 | 8 | 21.5 | 18.5 | FUSED | SIGLIP |
| DECODER AND QF HIDDEN SIZE 128 | 28.4 | 8.4 | 22.4 | 18.8 | FUSED | SIGLIP |
| LAYERNORM EPS 1E-5 | 30 | 8.3 | 22.3 | 19.6 | FUSED | SIGLIP |
| DROPOUT PROBABILITY 0.1 | 29.7 | 8.2 | 20.9 | 19.8 | FUSED | SIGLIP |
| ADAMW WITH DEFAULT EPS AND BETA2 | 29.5 | 8 | 21.1 | 19.1 | FUSED | SIGLIP |
| FINAL MODEL: LR 0.001 | 29.3 | 8.5 | 22.8 | 19.1 | FUSED | SIGLIP |
| FINAL MODEL: LR 0.002 | 23.1 | 7.8 | 15.8 | 18.8 | FUSED | SIGLIP |
| FINAL MODEL: LR 0.0005 SEED0 | 32.8 | 8.4 | 23.7 | 20.1 | FUSED | SIGLIP |
| FINAL MODEL: LR 0.0001 | 31.3 | 8.3 | 25.1 | 19.4 | FUSED | SIGLIP |
| FINAL MODEL: LR 0.0003 | 33.8 | 8.5 | 26.2 | 20.1 | FUSED | SIGLIP |
| FINAL MODEL: LR 0.0006 | 23.6 | 8.4 | 19 | 18.6 | FUSED | SIGLIP |

**Contributions**. In (Awad et al., 2023) the authors demonstrated how a deep learning module which encode a sequence of image-and-text items using diverse representations composed on several modalities, across time steps, and across pooling methods, obtained impressive results in sponsored search and recommendations. Inspired by the ADPM in (Awad et al., 2023) and using tricks to train vision transformers in (Dolev et al., 2023) and (Karamcheti et al., 2024), a smarter VLM is built. In this article, we make the following **contributions** on the VLM reasoning axis:

- Introduce a novel multimodal QF-layer to learn a hidden representation from the vision and language modalities.

- Improve the MLP decoders in (Cherian et al., 2022) through GELU activations, residual connections, and layer normalization.

- Improve the sequence decoder by replacing the LSTM with a GRU.

- Strengthen the vision modality by learning an adaptive visual representation on top of two fused vision backbones: SigLIP (Zhai et al., 2023) and DinoV2 (Oquab et al., 2023) similarly to (Karamcheti et al., 2024). In this way, the model makes better use of the puzzle's image.

- Strengthen the text-vision alignment by using a frozen SigLIP language encoder together with the vision modality which includes the SigLIP vision backbone. The pretrained text encoder does not overpower the visual signal as much as an LLM as seen in (Tong et al., 2024; Zhang et al., 2024b).

- Furthermore, the smarter VLM reasoner includes a composite hidden representation through the concatenation of language-only representations, an adaptive image-only representation learned on top of the fused frozen foundation backbones, and the QF multimodal layer representation which includes a language-vision cross-attention sublayer. Ablation studies in Section 4 show that the QF layer is essential to the smarter VLM reasoner. The use of cross-attention improves the ability of the reasoner to make use of the puzzle's visual cues.

- These model improvements lead to up to $48\%$ accuracy gain across several of the meta reasoning skills measured by the challenging SMART task.

## 3. Methodology

We formalize the problem as supervised learning with **classification loss.** For each image-question instance, **we predict the probability of one of five answer** options. When the

*Table 3.* QF Ablations. Validation Accuracy per Skill Class (counting, math, logic, path) per Architectural, Optimization and Hyperparameter Choices. The fused vision encoder is DinoV2+SigLIP and the text encoder is SigLIP. From CometML multimodalAI.

| CHOICES | COUNTING | MATH | LOGIC | PATH |
|---|---|---|---|---|
| SMARTEST VLM | 33.8 | 8.5 | 26.2 | 20.1 |
| 1 MHA HEADS | 29.7 | 8.2 | 23.1 | 19.4 |
| 3 MHA HEADS | 34.2 | 8.6 | 25.8 | 19.9 |
| 4 MHA HEADS | 32.9 | 8.7 | 25.6 | 20.1 |
| 8 MHA HEADS | 33.1 | 8.4 | 26.9 | 20.3 |
| QF INTERMEDIATE SIZE 128 | 33.1 | 8.6 | 25.6 | 19.8 |
| QF INTERMEDIATE SIZE 512 | 33.4 | 8.1 | 27.3 | 19.9 |
| QF INTERMEDIATE SIZE 768 | 33.4 | 8.7 | 25.1 | 19.3 |
| QF INTERMEDIATE RELU | 32.8 | 8.7 | 26.7 | 19.8 |
| QF INTERMEDIATE SILU | 33.2 | 8.7 | 26.5 | 19.6 |
| COMPOSITE: NO QF LAYER | 32.8 | 8.5 | 23.8 | 19.7 |
| COMPOSITE: QF ONLY | 32.2 | 8.0 | 26.3 | 18.8 |
| COMPOSITE: QF AND VISION ONLY | 33.7 | 8.7 | 24.8 | 20.4 |
| COMPOSITE: QF AND LANGUAGE ONLY | 33.6 | 8.9 | 24.9 | 19.4 |
| NO residual connection in QF intermediate | 32 | 8.4 | 26.4 | 19 |
| DROPOUT 0 IN QF LAYER | 33 | 8.2 | 26.9 | 19.6 |
| DROPOUT 0.1 IN QF LAYER | 30.3 | 8.1 | 25.5 | 19 |

*Table 4.* QF Layer Ablations. Validation Accuracy per Skill Class (algebra, measure, spatial, pattern) per Architectural, Optimization and Hyperparameter Choices. The fused vision encoder is DinoV2+SigLIP and the text encoder is SigLIP. From CometML multimodalAI.

| CHOICES | ALGEBRA | MEASURE | SPATIAL | PATTERN |
|---|---|---|---|---|
| SMARTEST VLM | 11.2 | 10.4 | 26.8 | 27 |
| 1 MHA HEADS | 10.5 | 10.8 | 23.2 | 22.7 |
| 3 MHA HEADS | 11.1 | 11.3 | 26.8 | 27.0 |
| 4 MHA HEADS | 11.2 | 10.6 | 27.8 | 26.6 |
| 8 MHA HEADS | 11.6 | 10.4 | 27.9 | 25.8 |
| QF INTERMEDIATE SIZE 128 | 11.1 | 10.2 | 26.9 | 27.1 |
| QF INTERMEDIATE SIZE 512 | 11.3 | 10.4 | 27.8 | 26.4 |
| QF INTERMEDIATE SIZE 768 | 11.3 | 11.5 | 27 | 26.7 |
| QF INTERMEDIATE RELU | 10.8 | 9.9 | 28.1 | 25.5 |
| QF INTERMEDIATE SILU | 10.9 | 9.5 | 27.4 | 26 |
| COMPOSITE: NO QF | 11.5 | 10.3 | 25.6 | 25.4 |
| COMPOSITE: QF ONLY | 11.3 | 10.7 | 27.4 | 25.7 |
| COMPOSITE: QF AND VISION ONLY | 11.3 | 11.5 | 27.3 | 27.4 |
| COMPOSITE: QF AND LANGUAGE ONLY | 11.1 | 10.9 | 28.2 | 25.9 |
| NO residual connection in QF intermediate | 11.2 | 10.6 | 26.9 | 26.7 |
| DROPOUT 0 IN QF LAYER | 11.1 | 9.6 | 27.7 | 27.4 |
| DROPOUT 0.1 IN QF LAYER | 11.1 | 10.1 | 25.7 | 23.7 |

options are in the form of a sequence, a decoder module decodes the answer sequence first, and then the answer is translated to one of the $\{A, B, C, D, E\}$ multiple choice options. In this article, the decoder is a recurrent neural network (Cho et al., 2014). Furthermore, we focus on training deep learning architectures from scratch for the SMART task with inputs from diverse pretrained **frozen** backbones. We focus the investigation on the eight skill classes counting, (**counting, math, logic, algebra, path, pattern, measure, spatial**) rather than individual puzzle groups, since these reasoning skills are of more general interest across domains and trademarks of intelligence. In (Chen et al., 2019) authors demonstrate that strong pretrained backbones can perform without meta-learning, so we do not employ metalearning as in (Cherian et al., 2022). The accuracy metric calculated on the validation set is used to tune the models and evaluate method success, and the accuracy for the five-class classification is evaluated on the test set. The accuracy is calculated overall, and more interestingly, broken down by reasoning skill class (counting, math, etc.).

We derive a multimodal representation through a novel layer,

the QF layer, inspired by the ADPM in adsFormers (Awad et al., 2023), the QFormer in (Li et al., 2023b; Zhu et al., 2024) and VilBERT in (Lu et al., 2019). More recently, (Karamcheti et al., 2024) and (Tong et al., 2024) combine multiple image representations to leverage diversity of signal in vision-language models, in a similar vein to the ADPM. Additionally, inspired from (Cherian et al., 2022), we learn an adaptive image representation on top of the fused frozen backbone. Moreover, we directly include the frozen SigLIP text encoder to encoder the question tokens. The choice of the SigLIP text encoder is two-fold: firstly, the alignment with the SigLIP vision encoder; secondly, to tame the language power by not employing a very large language model (LLM) such as (Gao et al., 2023), which standard VLM such as BLIP (Li et al., 2023b) or Llava (Liu et al., 2023) employ. In (Tong et al., 2024) we see that visual grounding is lacking in large multimodal models. Furthermore, in (Cherian et al., 2022) the visual signal is quite important, the accuracy loss is more when removing the image rather than the text question, so the visual signal needs to be protected as it is critical for the reasoning skill necessary to solve the puzzles.

The QF layer representation learned from the image-question input is concatenated to the average pooled text representation and the puzzle-specific adaptive image representation from the fused image encoders. As depicted in Figure 1, the resulting composite representation, denoted $compositeR$, takes as input three component representations, $r_1, r_2$, and $r_3$, defined as follows.

- Text representation $r_3$ is an average pooled encoding of the question sequence of max length 110 tokens. Each token is first encoded using the frozen SigLIP text model into a representation of size 768. Then

$$r_3 = AveragePooling([h_1, h_2, ..., h_{110}]). \quad (1)$$

- An image representation $r_1$ from the puzzle-specific image encoder block of dimension 128 seen in 1. The dimension is a hyperparameter selected via optimization. The image encoder consists of two feed forward layers with a GELU unit, with separate weights for each puzzle head, for the 101 separate puzzle groups (e.g. one loss calculated per puzzle-group). Each encoder takes as input the image representation from the two fused pretrained vision backbones, DinoV2 (Oquab et al., 2023) and SigLIP (Zhai et al., 2023), each of dimension 768. Specifically, for an image $X$, $r_1$ is

$$r_1 = FC_{1i}(GELU(FC_{2i}(y))), \quad (2)$$
$$y = Concat([Dino(x), SigLIP(x)]) \quad (3)$$

for $i \in \{1, \ldots, 101\}$, a distinct puzzle group.

- A QF representation, $r_2$, is produced by the QF layer which takes as input the encoded image representation $r_1$ and the SigLIP-encoded sequence of text tokens (before average pooling). First, the SigLIP frozen language backbone encodes the 110-long question text sequence. Then, the QF layer passes the text sequence through a multi-head self-attention block (Vaswani et al., 2017). The resulting hidden representation is fed to a cross-attention layer as query, with keys and values coming from the adaptive image encoder representation, marginally inspired from the QFormer in (Li et al., 2023b) and VilBERT (Lu et al., 2019) but with multiple differences. Distinctly from these works, in our case the image encoder in the cross-attention sublayer is a per-puzzle group adaptive representation learned on top of the frozen fused concatenation of DinoV2 and SigLIP vision backbones. Finally, an intermediate stack of fully connected layers, with residual connections (He et al., 2016), dropout (Srivastava et al., 2014), and layer normalisation (Ba et al., 2016) produces the

QF text-and-vision multimodal representation. Specifically,

$$r_2 = LayerNorm(x + Drop(FC(GELU(FC(x))))) \quad (4)$$
$$X = MHCrossA(MHA([h_1, h_2, ..., h_{110}]), r_1) \quad (5)$$

Finally, the composite QVFusion layer aggregates these distinct representations (text-only, vision-only, and text-and-vision QF multimodal) via concatenation producing the composite representation $CompositeR \in \mathbb{R}^{2*768+128}$, and then passing it through a two-layer feed forward module with Gaussian error linear units (Hendrycks & Gimpel, 2016) in between , before being read by the puzzle specific decoder.

The composite representation is

$$CompositeR = CLayer([r_1, r_2, r_3]) \quad (6)$$
$$= LayerNorm(Concat([r_1, r_2, r_3]). \quad (7)$$

The QVFusion layer in 1 is

$$QVFusion(y) = LayerNorm(GELU(y)) \quad (8)$$
$$y = FC(GELU(FC(compositeR))). \quad (9)$$

Finally, the decoder, which is either a stack of three fully connected layers separated by GELU activations, or a gated recurrent neural network (GRU) (Cho et al., 2014) for sequence-type answer puzzles, produces predictions fed to a cross-entropy loss. The introduction of GELU units with layer normalization boosts performance, as they do in many recent attention-based multimodal neural networks (Liu et al., 2023; Alayrac et al., 2022; Li et al., 2023b), by allowing for a smoother loss landscape than rectified linear units with batch normalization layers.

## 4. Experiments and Results

We train several **baselines** from (Cherian et al., 2022) and give results in Table 1. We chose to move forward with the frozen BERT+ResNet50 as baseline for two reasons: 1. Note that the numbers are extremely close between the frozen and unfrozen variants but the frozen variant does better on Math, a top skill of interest for this investigation; 2. The backbones are frozen which we favor in this article for a few reasons. The first reason is efficiency of training. Frozen backbones result in fewer parameters to update. Secondly, as noted in (Karamcheti et al., 2024), finetuning vision

backbones can deteriorate the performance of the vision-language model. Thirdly, keeping the backbone frozen affords a better comparison between models and the ability to reuse some hyperparameter settings such as batch size, number of epochs, or optimizer choice. If we choose to train the vision backbones, transformers and CNN-based ResNets typically need to employ different training choices. Next, we discuss experimental results on a subset of the SMART-101 training set.

**Challenges and limitations** for getting better models include compute requirements-to tune deep neural networks one needs to run many experiments. Large transformers are data hungry as well so one needs GPUs with large memories and disk space for extended periods of time. For efficiency of resource utilisation, we sampled the dataset and raised the bar on the model architectures rather than on data and compute. A dataset split of $train : val : test = 60 : 20 : 20$ and only 1000 out of the total 2000 question-image pairs per puzzle group are utilized for fast training and insight, with a total budget of three epochs of training. This results in 474 training batches, 158 validation and 158 test batches of size 128. The additional model layers described by Equations 1 through 9 result in 29,623,375 trainable parameters for the final best model. All the models were trained on one NVIDIA V100 GPU with 40Gb of memory, for 1-2 hours when training on the downsampled dataset. Experiments are tracked in CometML (Comet.com, 2021), the multimodalAI public project.

**Training and Evaluation**. Tables 2 and 6 (in the Appendix) show results from experiments ablating architecture, optimization, and hyperparameter decisions toward the final model. To avoid an inefficient combinatorial explosion of hyperparameter choices, the authors' deep learning experience guided the experimentation process in a stepwise fashion, aided by learning curve visualisations in CometML. Watching training and validation curves is an intimate part of the deep learning development process. In Figure 3 we can see training loss nicely descending over the training steps across the three epochs modulated by five different learning rates with a cosine scheduler (Loshchilov & Hutter, 2016) which adapts the learning rate based on the step number. Note how the large 0.002 learning rate (orange) impacts learning negatively in the strongest way. Noticeable bumps in curves depend on the scheduler's change points. **Reproducibility**. All the experiments are run with seed set to zero for the sake of reproducibility. Furthermore, by tracking machine learning training and evaluation experiments with CometML, all the hyperparameters for a given experiments are recorded, loss and accuracy curves and metrics, as well as running code which can be checked out from the linked repository. All results are fully reproducible.

**Final results**. As can be seen in Table 5, the best trained

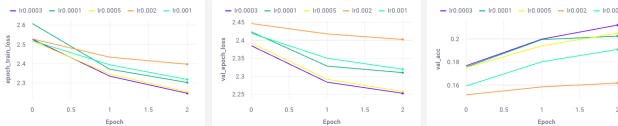

*Figure 3.* Epoch Train Loss, Validation Loss, and Validation Accuracy for five different learning rates. From CometML multimodalAI.

models display massive gains in accuracy (eg. **+48% gain over the baseline in the counting skill**) across reasoning skills. Recall that the baseline was chosen as the strongest in the math skill. Truly so, the baseline was hardest to beat in math vs. other skills. Tables 2 and 6 (in the Appendix) list the hyperparameter setting and other training choices experimented with to arrive at the best results in Table 5. In summary, the smartest VLM reasoner was trained using the following deep learning choices, inspired as a starting point from results in (Awad et al., 2023; Dolev et al., 2023; Roberts, 2019; Olteanu Roberts, 2021):

- Adam optimizer with decoupled weight decay from (Loshchilov & Hutter, 2018) with weight decay of 0.2, $eps = 1e - 8$, and $beta2 = 0.98$.

- Cosine learning rate scheduler (Loshchilov & Hutter, 2016) with ten warmup steps, using the implementation in the HuggingFace repository (Wolf et al., 2019).

- Clipping the gradient norm to no more than one to avoid explosions.

- Layer normalization throughout the architecture modules with $eps = 1e - 6$ for better learning and generalization.

- A SigLIP frozen language backbone and fused DinoV2 and SigLIP vision backbone.

- A composite hidden representation with a QF layer with two attention heads, concatenated with text-only and adaptive vision-only representations.

- Dropout probability of 0.2 anywhere dropout is used.

- Employing Gaussian error linear units instead of rectified linear units in the decoder leads to improvements across the eight skills. The MLP decoder is akin to SigLIP's MLP block (a stack of feed-forward layers with GELU activation).

- Hidden representation sizes of 128 (for instance for the adaptive image representation), except the hidden size within the GRU which is 256. Note that we learn

*Table 5.* Test set skill class accuracy for top models and comparison to the baseline in first row (percentage change). From CometML multimodalAI.

| NEURAL NET | COUNTING | MATH | LOGIC | PATH | ALGEBRA | MEASURE | SPATIAL | PATTERN | OVERALL |
|---|---|---|---|---|---|---|---|---|---|
| BERT+RESNET50 | 23.4(-) | 9.6(-) | 17.9(-) | 17.5(-) | 10.5(-) | 9.9(-) | 25.8(-) | 20.3(-) | 17.1(-) |
| SMARTERVLM LR0.001 | 29.0(+24%) | 9.9 (+3%) | 21.2 (+18%) | 17.9(+2%) | 10.8 (+3%) | 11.1 (+12%) | 23.2 (-10%) | 25.7 (+27%) | 19.12 (+12%) |
| SMARTERVLM LR0.0005 | 32.9(+41%) | 10.0(+4%) | 22.8(+27%) | 19.5(+11%) | 11.2(+7%) | **11.6(+17%)** | 26.3(+2%) | 25.8(+27%) | 20.86(+22%) |
| *SmartestVLM lr0.0003* | **34.7(+48%)** | 9.5(-1%) | **25.7(+44%)** | 19.5(+11%) | 11.3(+8%) | 11.1(+12%) | **26.7(+3%)** | **27.4(+35%)** | **21.59(+26%)** |
| SMARTERVLM (NO QF) | 32.3(+38%) | **10.3(+7%)** | 23.3(+30%) | 18.8(+7%) | 10.0(-5%) | 10.1(+2%) | 25.8 (+0%) | 23.6(+16%) | 20.14(+18%) |

an adaptive per-puzzle group visual representation by using a fully connected layer on top of the fused vision backbone.

- GRU decoder for problems with sequence answer, as they are easier to train than LSTMs.

A vital insight arose through the training process. In Figure 4 notice how the eight skill sets have different training dynamics and respond differently to learning rate choices, as well as to the cosine scheduler's learning rate decision throughout the training steps. This is something commonly seen in multitask learning (Caruana, 1997; Dolev et al., 2023), and sparks one of our recommendations in the future work section. All experiments are run with seed 0 for the sake of reproducibility but we also evaluated the test accuracy standard deviation across a few seeds (0, 42, 7) with mean overall test accuracy 20.8 ($\sigma = 0.16$), math skill mean accuracy of 9.73 ($\sigma = 0.38$), and pattern mean accuracy 25.2 ($\sigma = 0.53$). The other skills having similar variability. As expected, we see more variability on smaller individual skill class sets in terms of how many puzzles are in that specific skill category.

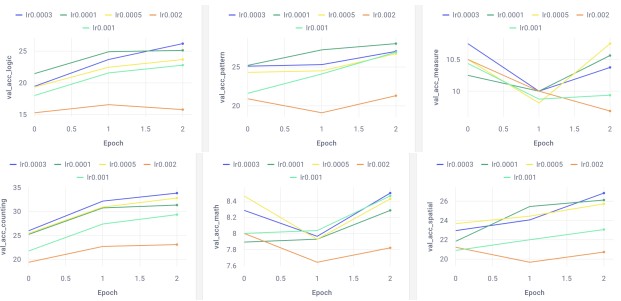

*Figure 4.* Validation accuracy curves per skill class (counting, math, spatial, logic, pattern, measure) for five different learning rates.

**QF layer ablations.** The QF layer employs a multihead self-attention (MHA) sublayer using the SigLip language encoder representations and a cross-attention sublayer which uses the text hidden MHA representations as queries and the adaptive image representations as keys and values. There-

fore the hidden sizes in the MHA QF subnetworks are fixed to the representation sizes of the language and vision encoder representations. There is flexibility on the intermediate subnetwork composition as well as the number of heads, addition of normalization, dropout, and residual layers. We have performed extensive ablation experiments to understand the impact of these different choices for the QF layer, and how the learned QF representation performs as part of the QVFusion composite representation where it is concatenated with vision and language-only representations.

**Constituency of composite representation ablation results.** From Tables 3 and 4 we can see that the inclusion of the QF layer representation together with the vision-only, language-only, both vision and language or QF representation-only improves accuracy on all skill sets. Intuitively, the model can better use the word-language cross-signals to make sense of the puzzles. Interestingly some skill sets benefit from visual cues more, where only QF and vision representations are included in the composite representation and not language (pattern and measure) while other skills (math, spatial) benefit from using QF and language representations (no image representation concatenated in the composite QVFusion), which perhaps is an indication that in some math puzzles the model cannot make good use of the diagram/image as mentioned in (Zhang et al., 2024b). The insight here is that the model can still make sense from the cross-signal from text-image encoded through the QF layer which includes a learned cross-attention sublayer with language as queries and the adaptive vision representation (learned on top of the fused frozen vision encoders) as keys and values. Intuitively this makes sense because humans will rely on the visual cues more for some type of problems and more on the verbal cues for others. Furthermore, one theory of learning postulates that some humans are more "visual learners" while others are more "auditory" learners, relying more on speech and language (Pennsylvania Higher Education Assistance Agency).

**Activation function used inside the QF layer.** We ablated the activation function to use the self-gated activation (Ramachandran et al., 2017) instead of GELU, as well as ReLU. Several of the large language models, such as (Touvron et al., 2023; Jiang et al., 2024), use the SiLU activation, which motivated our choice. We found that GELU works best on

most skills, except on math-type puzzles, where both ReLU and SiLU work better. Based on these results only we do not have an intuition of why this might happen. We will perform additional future experiments on a larger dataset for deeper understanding.

**QF intermediate sublayer ablations**. The QF layer includes a multihead attention (MHA) sublayer which takes the frozen text representations as input, a cross-attention layer which uses the MHA hidden representations as queries and the adaptive image representations as keys and values, and an intermediate final stack consisting of two fully connected layers and a residual connection with dropout and layer normalization. Exclusion of the residual connection (together with dropout and layer normalization) had a detrimental impact across skills, confirming the importance of residuals and regularization techniques for learning in deep transformers and for generalization. Furthermore, ablation on the dropout level, confirms better generalization with more regularization from dropout. The ablation of the intermediate layer sizing had mixed results across reasoning skills and we proceeded with the symmetry of using the hidden size used elsewhere throughout the smartest reasoner architecture (256). The QF layer includes two types of multihead attention (language only self-attention and language-vision cross-attention) which share the number of heads. An ablation on number of heads (1, 2, 4, 8) showed mixed results across skills with very similar results for average accuracy (except one head, which underperforms all the other choices) and is worth experimenting with further on larger datasets in future work.

**Development process and scaling side note**. For a deep understanding of the behavior of the models with various architectural and hyperparameter choices, in the development phase, we started with training and evaluation on a very small subset of data for quick insight and iteration: only 20 questions per puzzle, with a batch size of 16, on a split ratio of $train : val : test = 40 : 20 : 40$. The experimental results are available in Tables 7 and 8 in the Appendix and were tracked with CometML (Comet.com, 2021) and publicly accessible at vlm-reasoners.

## 5. Discussion and Future Work

In this article, we show how deep learning architectural innovations as well as hyperparameter and training choices led to improvement in model performance on the SMART reasoning task. Multimodal transformer-like architectures, deep learning representations, and stronger visual grounding led to improvements in eight fundamental reasoning skills of vision language models.

**Future work**. Considering the different learning dynamics for the eight skill classes, a **multitask learning ap-**proach (Caruana, 1997; Lu et al., 2022; Dolev et al., 2023) with eight tasks may afford modulating the impact of eight weighted losses to account for the different dynamics. A mixture-of-experts approach (Zhao et al., 2019) within the multitask learning framework could further help. Further experimentation with other custom layers similar to the QF layer in this article which facilitates better synergies across modalities may spark further improvements across fundamental reasoning skills. Experimenting with other simple or composite general purpose **backbones across modalities**, deeper or wider, is another potential avenue, based on improvements seen in this work due to the fused DinoV2+SigLIP.

Efficient training techniques employing **compression** (Dettmers et al., 2024), autodiff variations (Roberts & Roberts, 2020), or variations of multimodal transformers' efficient training (Liu et al., 2024b) may facilitate access to larger model sizes. In this article, visual and text representations are combined through concatenation. Experiments with an approach similar to (Ramrakhya et al., 2024), where learned representations on the decoder path take frozen visual features as inputs and are conditioned on text embeddings via an **element-wise product**, show faster learning and better performance in the context of Math AI, according to initial experiments. Decoder-only architectures (Radford et al., 2019) took generative modeling by storm, and exploring **other decoding architectures** for the VLM reasoner instead of recurrent neural networks, such as Perceiver-inspired layers (Alayrac et al., 2022), or convolutional neural networks(He et al., 2016; Roberts, 2019), may show interesting results. Furthermore, utilizing a frozen **text encoder pretrained to outperform in mathematical reasoning** and for efficiency, such as Mistral in (Jiang et al., 2023), or Mixtral in (Jiang et al., 2024), in conjunction with the strengthened vision encoders and cross-attention layers introduced in this article, is a further avenue worth exploring for next generation VLM reasoners.

Additionally, in (Hu et al., 2024) LLMs are instructed to follow rules for general problem solving, instead of relying on the cases already seen in training. We expect that in the multimodal smart puzzle solving case, where we have multiple generated instances of each unique root puzzle, a similar investigation may improve generalization to unseen root puzzles. In fact, based on the progress in the quality of visual representations generated for multimodal mathematical knowledge (Wu et al., 2024), more problems could be automatically generated using machine generated representations based on models, for further **data augmentation** to improve learning and generalization. While we deep dive into the SMART benchmark, several other multimodal evaluation benchmarks were recently published in the AI for Math space, as mentioned in the literature review section, and our architecture can be applied to any of them, which

we leave for future work. Finally, other Math AI tasks, such as theorem proving and other tasks mentioned in the excellent survey article (Lu et al., 2022), can benefit from our methodology.

## Impact Statement

This paper presents work whose goal is to advance the field of Machine Learning and Math AI. There are many potential societal consequences of our work, none which we feel must be specifically highlighted here.

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

*Table 6.* Follow up to Table 2 from main text. Validation Accuracy per Skill Class (algebra, measure, spatial, pattern) per Architectural, Optimization and Hyperparameter Choices. The fused vision encoder is DinoV2+SigLIP. From CometML multimodalAI.

| CHOICES | ALGEBRA | MEASURE | SPATIAL | PATTERN | VISION | LANGUAGE |
|---|---|---|---|---|---|---|
| BASELINE: RESNET50+MBERT | 10.3 | 10.2 | 23.8 | 20.8 | RESNET50 | MBERT |
| BASELINE: RESNET50+BERT | 10.7 | 10.3 | 24.8 | 20.5 | RESNET50 | BERT |
| LSTM DECODER SIGLIP VISION | 10 | 9.1 | 22.6 | 21.4 | SIGLIP | SIGLIP |
| LSTM DECODER WITH FUSED VISION | 11.2 | 9.2 | 23.3 | 25.1 | FUSED | SIGLIP |
| NON-ADAPTIVE IMAGE REPRESENTATION | 10.3 | 10.2 | 23.4 | 22.1 | FUSED | SIGLIP |
| EXTRA RESIDUAL CONNECTION IN MLP DECODER | 9.3 | 10.4 | 21.6 | 20.3 | FUSED | SIGLIP |
| WARMUP STEPS 0 | 10.1 | 10.8 | 22.8 | 26.6 | FUSED | SIGLIP |
| WARMUP STEPS 0.06 PERCENT | 9.6 | 9.8 | 22.9 | 25.1 | FUSED | SIGLIP |
| WARMUP STEPS 0.01 PERCENT NO EXTRA RESIDUALS | 9.8 | 9.8 | 21.1 | 26.5 | FUSED | SIGLIP |
| 10 WARMUP STEPS | 9.5 | 9.7 | 23.2 | 19.8 | FUSED | SIGLIP |
| BATCH SIZE 64 | 10.1 | 10.8 | 22.3 | 26 | FUSED | SIGLIP |
| ADAPTIVE IMAGE REPR SIZE 256 | 10.3 | 10.6 | 23.8 | 26.3 | FUSED | SIGLIP |
| DECODER AND QF HIDDEN SIZE 128 | 10.6 | 10.2 | 25.9 | 23.3 | FUSED | SIGLIP |
| LAYERNORM EPS 1E-5 | 9.9 | 9.8 | 22.8 | 26.1 | FUSED | SIGLIP |
| DROPOUT PROBABILITY 0.1 | 10 | 9.6 | 23.1 | 26.2 | FUSED | SIGLIP |
| ADAMW WITH DEFAULT EPS AND BETA2 | 10.7 | 9.7 | 23.2 | 26.8 | FUSED | SIGLIP |
| FINAL MODEL: LR 0.001 | 10.3 | 9.9 | 23.1 | 26.9 | FUSED | SIGLIP |
| FINAL MODEL: LR 0.0005 SEED0 | 10.4 | 10.8 | 25.7 | 26.7 | FUSED | SIGLIP |
| FINAL MODEL: LR 0.0001 | 10.4 | 10.6 | 26.1 | 28 | FUSED | SIGLIP |
| FINAL MODEL: LR 0.0003 | 11.2 | 10.4 | 26.8 | 27 | FUSED | SIGLIP |
| FINAL MODEL: LR 0.0006 | 9.5 | 10.6 | 23.9 | 22.4 | FUSED | SIGLIP |
| FINAL MODEL: LR 0.002 | 10.2 | 9.7 | 20.7 | 21.3 | FUSED | SIGLIP |

## A. Further Experimental Results.

Results in Table 6 correspond to the algebra, measure, spatial, and pattern reasoning skills for the ablation experiments presented in the main text and a continuation to the results in Table 2 which included accuracy results for the counting, math, logic, and path skills.

Furthermore, experimental results in Tables 7 and 8 from vlm-reasoners were obtained on a small subset of the SMART-101 dataset and can be reproduced using the end-to-end code github.com/smarter-vlm/smarter for the same hyperparameter and architectural choices. Table 7 gives results for the first four skills, counting, math, logic, and path, and Table 8 gives results for the remaining four skills, algebra, measure, spatial, and pattern.

*Table 7.* Small Dataset Experimental Runs: Architecture and Hyperparameter Choices Impact on Skill Class Accuracy. From vlm-reasoners.

| CHOICE | COUNTING | MATH | LOGIC | PATH | QF_HEADS | PDROP | WD | TEXT | VISION |
|---|---|---|---|---|---|---|---|---|---|
| LR0.005 | 13.5 | 8.9 | 5.6 | 16.7 | 2 | 0.2 | 0.2 | SIGLIP | FUSED |
| LR0.0001 | 18.3 | 10.7 | 5.6 | 14.6 | 2 | 0.2 | 0.2 | SIGLIP | FUSED |
| DROPOUT 0.1 | 13.5 | 10.7 | 2.8 | 18.8 | 2 | 0.1 | 0.2 | SIGLIP | FUSED |
| EXTRA RESIDUAL IN DECODER | 14.4 | 7.1 | 5.6 | 16.7 | 2 | 0.2 | 0.2 | SIGLIP | FUSED |
| SIGLIP VISION | 12.5 | 7.1 | 5.6 | 16.7 | 2 | 0.2 | 0.2 | SIGLIP | SIGLIP |
| ADAPTIVE VISUAL REPR SIZE 64 | 13.5 | 7.1 | 5.6 | 16.7 | 2 | 0.2 | 0.2 | SIGLIP | FUSED |
| ADAPTIVE VISUAL REPR SIZE 64 | 18.3 | 8.9 | 5.6 | 16.7 | 2 | 0.2 | 0.2 | SIGLIP | FUSED |
| NO QF LAYER | 12.5 | 5.4 | 5.6 | 18.8 | 2 | 0.2 | 0.2 | SIGLIP | FUSED |
| COMPOSITE ALL | 15.4 | 10.7 | 5.6 | 18.8 | 2 | 0.2 | 0.2 | SIGLIP | FUSED |
| COMPOSITE NO TEXT | 16.3 | 7.1 | 5.6 | 14.6 | 2 | 0.2 | 0.2 | SIGLIP | FUSED |
| COMPOSITE NO IMAGE | 10.6 | 5.4 | 5.6 | 16.7 | 2 | 0.2 | 0.2 | SIGLIP | FUSED |
| VARIATION ON NON-ADAPTIVE IMAGE | 17.3 | 3.6 | 5.6 | 14.6 | 2 | 0.2 | 0.2 | SIGLIP | FUSED |
| NO GELU IN DECODER | 13.5 | 10.7 | 5.6 | 16.7 | 2 | 0.2 | 0.2 | SIGLIP | FUSED |
| ADD RESIDUAL BACK IN QF | 16.3 | 8.9 | 11.1 | 14.6 | 2 | 0.2 | 0.2 | SIGLIP | FUSED |
| NO RESIDUAL IN QF | 12.5 | 7.1 | 5.6 | 18.8 | 2 | 0.2 | 0.2 | SIGLIP | FUSED |
| QF INTERM. SIZE 128 | 17.3 | 5.4 | 5.6 | 16.7 | 2 | 0.2 | 0.2 | SIGLIP | FUSED |
| QF INTERM. SIZE 768 | 15.4 | 5.4 | 8.3 | 16.7 | 2 | 0.2 | 0.2 | SIGLIP | FUSED |
| COSINE SCHEDULER | 16.3 | 8.9 | 11.1 | 14.6 | 2 | 0.2 | 0.2 | SIGLIP | FUSED |
| NO COSINE SCHEDULER | 8.7 | 3.6 | 8.3 | 6.3 | 2 | 0.2 | 0.2 | SIGLIP | FUSED |
| RELU EVERYWHERE | 14.4 | 8.9 | 11.1 | 14.6 | 2 | 0.2 | 0.2 | SIGLIP | FUSED |
| QF 3 HEADS | 15.4 | 5.4 | 5.6 | 16.7 | 3 | 0.2 | 0.2 | SIGLIP | FUSED |
| QF 1 HEAD | 17.3 | 5.4 | 8.3 | 16.7 | 1 | 0.2 | 0.2 | SIGLIP | FUSED |
| LSTM DECODER (NOT GRU) | 14.4 | 8.9 | 5.6 | 16.7 | 2 | 0.2 | 0.2 | SIGLIP | FUSED |
| WD0 | 14.4 | 7.1 | 11.1 | 14.6 | 2 | 0.2 | 0 | SIGLIP | FUSED |
| WD0.05 | 14.4 | 7.1 | 11.1 | 14.6 | 2 | 0.2 | 0.05 | SIGLIP | FUSED |
| WD0.1 | 15.4 | 7.1 | 11.1 | 14.6 | 2 | 0.2 | 0.1 | SIGLIP | FUSED |
| WD0.2 | 15.4 | 7.1 | 11.1 | 14.6 | 2 | 0.2 | 0.2 | SIGLIP | FUSED |

*Table 8.* Small Dataset Experimental Runs: Architecture and Hyperparameter Choices Impact on Skill Class Accuracy. From vlm-reasoners.

| CHOICE | ALGEBRA | MEASURE | SPATIAL | PATTERN | QF_HEADS | PDROP | WD | TEXT | VISION |
|---|---|---|---|---|---|---|---|---|---|
| LR0.005 | 8.3 | 3.1 | 27.8 | 30 | 2 | 0.2 | 0.2 | SIGLIP | FUSED |
| LR0.0001 | 8.3 | 15.6 | 27.8 | 10 | 2 | 0.2 | 0.2 | SIGLIP | FUSED |
| DROPOUT 0.1 | 15 | 3.1 | 30.6 | 30 | 2 | 0.1 | 0.2 | SIGLIP | FUSED |
| EXTRA RESIDUAL IN DECODER | 8.3 | 9.4 | 27.8 | 25 | 2 | 0.2 | 0.2 | SIGLIP | FUSED |
| SIGLIP VISION | 10 | 15.6 | 33.3 | 15 | 2 | 0.2 | 0.2 | SIGLIP | SIGLIP |
| ADAPTIVE VISUAL REPR SIZE 64 | 11.7 | 6.3 | 27.8 | 25 | 2 | 0.2 | 0.2 | SIGLIP | FUSED |
| ADAPTIVE VISUAL REPR SIZE 64 | 10 | 9.4 | 30.6 | 20 | 2 | 0.2 | 0.2 | SIGLIP | FUSED |
| NO QF LAYER | 6.7 | 12.5 | 27.8 | 30 | 2 | 0.2 | 0.2 | SIGLIP | FUSED |
| COMPOSITE ALL | 11.7 | 3.1 | 30.6 | 30 | 2 | 0.2 | 0.2 | SIGLIP | FUSED |
| COMPOSITE NO TEXT | 6.7 | 12.5 | 27.8 | 25 | 2 | 0.2 | 0.2 | SIGLIP | FUSED |
| COMPOSITE NO IMAGE | 3.3 | 9.4 | 30.6 | 20 | 2 | 0.2 | 0.2 | SIGLIP | FUSED |
| VARIATION ON NON-ADAPTIVE IMAGE | 6.7 | 9.4 | 30.6 | 20 | 2 | 0.2 | 0.2 | SIGLIP | FUSED |
| NO GELU IN DECODER | 6.7 | 3.1 | 30.6 | 30 | 2 | 0.2 | 0.2 | SIGLIP | FUSED |
| ADD RESIDUAL BACK IN QF | 10 | 12.5 | 27.8 | 25 | 2 | 0.2 | 0.2 | SIGLIP | FUSED |
| NO RESIDUAL IN QF | 11.7 | 12.5 | 30.6 | 25 | 2 | 0.2 | 0.2 | SIGLIP | FUSED |
| QF INTERM. SIZE 128 | 6.7 | 12.5 | 25 | 25 | 2 | 0.2 | 0.2 | SIGLIP | FUSED |
| QF INTERM. SIZE 768 | 6.7 | 9.4 | 30.6 | 20 | 2 | 0.2 | 0.2 | SIGLIP | FUSED |
| COSINE SCHEDULER | 10 | 12.5 | 27.8 | 25 | 2 | 0.2 | 0.2 | SIGLIP | FUSED |
| NO COSINE SCHEDULER | 1.7 | 3.1 | 16.7 | 25 | 2 | 0.2 | 0.2 | SIGLIP | FUSED |
| RELU EVERYWHERE | 8.3 | 12.5 | 30.6 | 15 | 2 | 0.2 | 0.2 | SIGLIP | FUSED |
| QF 3 HEADS | 6.7 | 9.4 | 27.8 | 20 | 3 | 0.2 | 0.2 | SIGLIP | FUSED |
| QF 1 HEAD | 6.7 | 12.5 | 30.6 | 25 | 1 | 0.2 | 0.2 | SIGLIP | FUSED |
| LSTM DECODER (NOT GRU) | 6.7 | 12.5 | 27.8 | 25 | 2 | 0.2 | 0.2 | SIGLIP | FUSED |
| WD0 | 8.3 | 9.4 | 27.8 | 20 | 2 | 0.2 | 0 | SIGLIP | FUSED |
| WD0.05 | 8.3 | 9.4 | 27.8 | 20 | 2 | 0.2 | 0.05 | SIGLIP | FUSED |
| WD0.1 | 8.3 | 9.4 | 27.8 | 20 | 2 | 0.2 | 0.1 | SIGLIP | FUSED |
| WD0.2 | 8.3 | 9.4 | 25 | 25 | 2 | 0.2 | 0.2 | SIGLIP | FUSED |

