# OpenReview forum: "Smart Vision-Language Reasoners"
_ICML.cc/2024/Workshop/AI4MATH — ICML 2024 Workshop AI4MATH Poster_

### Official Review · Reviewer_ACuZ · 2024-06-10

**Rating:** 6
**Confidence:** 3

**Summary:**

The paper explores the concept of "Smart Vision-Language Reasoners" that leverage multimodal learning to improve mathematical reasoning and problem-solving abilities. The authors investigate the ability of vision-language models to reason along eight axes: math, counting, path, measure, logic, spatial, and pattern. The proposed smarterVLM model achieves significant improvements (up to 48% gain in accuracy) over baselines on the SMART task, a multimodal algorithmic reasoning task.

**Questions:**

* Can the authors provide a more detailed explanation of the QF layer and how it differs from existing multimodal fusion techniques?
* How did the authors select the specific vision and language models used in their approach, and what were the criteria for choosing these models?
* Can the authors provide a more detailed description of the training process, including the hyperparameter tuning and optimization methods used?
* How do the authors plan to address the limitations of the SMART task, such as its limited scope and potential biases?
* How do the authors plan to address the challenges of deploying their approach in real-world settings, such as scalability and robustness?

**Reasons To Accept:**

-- The paper presents a novel approach to multimodal reasoning, combining vision and language models to improve mathematical reasoning and problem-solving abilities. The authors' definition of Smart Vision-Language Reasoners and their application to the SMART task are innovative and original contributions to the field. The incorporation of a novel QF layer and the fusion of frozen pretrained backbones are also creative elements that enhance the paper's originality. However, the paper builds upon existing work in multimodal learning and vision-language models, which reduces its overall originality score.

-- The paper is well-written, and the authors provide a clear and concise explanation of their approach. The experimental design and methodology are sound, and the results are presented in a clear and easy-to-understand manner. The authors also provide a thorough discussion of related work, which helps to contextualize their contributions. The only area for improvement is in the presentation of the results, which could be enhanced with additional visualizations and more detailed analysis of the results.

-- The paper could benefit from a more detailed analysis of the results, including a more thorough discussion of the strengths and limitations of the approach.
-- The authors could provide more context for the SMART task and its relevance to real-world applications.
-- The paper could be enhanced with additional visualizations, such as diagrams or illustrations, to help illustrate the authors' approach and results.

**Reasons To Reject:**

-- While the paper presents a novel application of vision-language models to mathematical reasoning, the methodology itself is not particularly innovative. The use of transformers-like layers and frozen pretrained backbones is well-established in the literature .

-- The paper's evaluation is limited to a single benchmark (SMART task) and does not provide a comprehensive comparison with existing state-of-the-art methods. The authors should consider evaluating their approach on a broader range of tasks and comparing their results to existing methods.

-- The paper's results are presented in a concise manner, but lack a more in-depth analysis of the strengths and limitations of the approach. The authors could improve this by providing a more detailed breakdown of the results, including error analysis, ablation studies, and sensitivity analysis. This would provide a more comprehensive understanding of the approach's performance and limitations.

---

### Official Review · Reviewer_YGaR · 2024-06-12

**Rating:** 5
**Confidence:** 3

**Summary:**

The paper explores the enhancement of vision-language models (VLMs) for complex multimodal reasoning tasks, particularly in mathematics and problem-solving. Using the SMART task framework, the authors assess VLMs across eight reasoning skills. They employ frozen vision backbones (DinoV2, SigLIP) and introduce a novel QF layer to improve model performance. The paper compares various training strategies and hyperparameters, showing accuracy gains (up to 48%) in skills like counting and math. The research addresses challenges related to computational resources and proposes efficient solutions for data handling. Reproducibility is ensured through detailed experiment tracking and standardized settings.

**Questions:**

1. Could you provide more detailed results or an ablation study to isolate the impact of the QF layer on the overall performance of your model?
2. Are there specific examples or case studies that demonstrate the practical benefits of the proposed QF layer in real-world applications?

**Reasons To Accept:**

The authors presented a good evaluation of vision-language models (VLMs) for multimodal reasoning using the SMART task framework. Their creative use of frozen vision backbones and the new QF layer led to good performance boosts, with accuracy gains up to 48%. The reviewer also appreciates the effort of reproducibility the authors have put into the work.

**Reasons To Reject:**

1. The writing could be improved for clarity, as some sections are hard to follow and could benefit from more straightforward language. The reviewer can see that a lot of work went into the paper, but the key messages could be highlighted better to make the main contributions stand out.

2. The effectiveness of the proposed QF layer isn't clearly demonstrated; the paper lacks an ablation study to isolate the impact of this specific component. This makes it difficult to assess how much the QF layer contributes to the overall performance improvements. Including such an analysis would greatly strengthen the claims made about the QF layer's effectiveness.

---

### Meta-Review · Area_Chair_drdm · 2024-06-13

**Recommendation:** Accept (Poster)
**Confidence:** 4

**Metareview:**

The paper evaluates the effectiveness of large vision-language models (VLMs) in solving complex multimodal reasoning tasks. The paper is novel in combining the frozen vision backbones with the new QF layer and in evaluation with the SMART task framework. The author could further improve the paper by providing more details and ablation on the QF layer and giving a more in-depth analysis.

---

### Decision · Program_Chairs · 2024-06-13

Accept (Poster)